# Effect of dietary omega-3 fatty acid supplementation on frailty-related phenotypes in older adults: a systematic review and meta-analysis protocol

Joanne Stocks,[1,2,3] Ana M Valdes[1,2,3]

[1]NIHR Nottingham BRC, Nottingham, UK
[2]Arthritis Research UK Pain Centre, University of Nottingham, Nottingham, UK
[3]Division of Rheumatology, Orthopaedics and Dermatology, School of Medicine, University of Nottingham, Nottingham, UK

**Correspondence to**
Dr Joanne Stocks;
joanne.stocks@nottingham.ac.uk

## ABSTRACT

**Introduction** The beneficial effect of dietary omega-3 supplementation in younger adults or older people with acute or chronic disease is established. Knowledge is now needed about the effect in medically stable older people. The objective of this study is to examine and assess the evidence for a role of dietary omega-3 polyunsaturated fatty acid (PUFA) supplementation in older adults on (1) muscle mass and muscle strength, (2) inflammatory biomarkers and (3) physical activity.

**Methods and analysis** A systematic review and data synthesis will be conducted of randomised controlled trials in older people not recruited for any given disease diagnosis. Placebo-controlled studies reporting interventions involving dietary supplementation of omega-3 PUFAs, eicosapentaenoic acid and docosahexaenoic acid will be included. Outcomes must include changes from baseline to last available follow-up for one or more of the following: muscle mass, inflammatory biomarkers, physical activity, walking speed, weight change, hand grip strength or muscle strength. Once the search strategy has been carried out, two independent researchers will assess relevant papers for eligibility. Articles up until 31 December 2017 in any language will be included. We will provide a narrative synthesis of the findings from the included studies. Studies will be grouped for meta-analysis according to the outcome(s) provided. Where studies have used the same type of intervention, with the same outcome measure, we will pool the results using a random effects meta-analysis, with standardised mean differences for continuous outcomes and risk ratios for binary outcomes, and calculate 95% CI and two-sided p values for each outcome.

**Ethics and dissemination** No research ethics approval is required for this systematic review as no confidential patient data will be used. The results of this systematic review will be disseminated through publication in an open-access peer-reviewed journal and through conference presentations.

**PROSPERO registration number** CRD42017080240.

## Strengths and limitations of this study

► This will be the first study to systematically examine the effect of dietary omega-3 fatty acid supplementation on frailty-related phenotypes in older adults not selected for specific chronic or acute conditions.

► An important strength of the study is the focus on both functional (walking speed, grip strength, get up and go) and inflammatory outcomes (cytokine level) outcomes that will allow to put in context the effect size and direction of effect of omega-3 supplementation and to prioritise outcomes for future RCTs.

► Results will also help to inform future guidelines on dietary supplementation for older adults.

► Limitations may include issues of poor reporting affecting risk of bias assessment and confidence in results.

## INTRODUCTION

According to the United Nations, the number of people aged over 60 years will double globally from 962 million in 2017 to 2.1 billion in 2050.[1] In Europe, the proportion of the population aged over 60 years is projected to reach 35% by 2050.[1] It is, therefore, a global priority to ensure that this ageing population remains independent. A key element of maintaining independence in older adults is the preservation of mobility along with muscle mass and strength.

Muscle mass decline is one of the hallmarks of ageing and, from age 40 years, muscle mass begins to decrease, with an annual decline in functional capacity of up to 3% per year after age 60 years.[2] A key gerontological concept linked to musculoskeletal ageing is frailty.[3] The commonly acknowledged characteristics include unintentional weight loss, self-reported exhaustion, weakness (grip strength), slow walking speed and low physical activity.[4] This complex phenomenon is highly correlated with loss of mobility along with progressive loss of skeletal muscle strength (dynapenia), mass and function (sarcopenia)[5] and results in a reduced quality of life and is a major public health concern.[6]

The prevalence of frailty also increases with age, and along with sarcopenia is associated

with serious adverse outcomes, including falls, hospitalisation and mortality.[7] There is consensus of an inflammatory contribution to frailty. Striking differences in the levels of proinflammatory cytokines between frail and non-frail elderly have been reported[8] and predict higher mortality.[9]

A role for nutritional determinants of frailty has been proposed,[10] and a number of lifestyle interventions have been investigated with regards to frailty, including exercise and increased protein intake.[11]

Recent work has also begun to investigate such interventions to prevent or diminish muscle loss in medical settings, including the supplementation of leucine,[12] vitamin D[13] and fish-derived omega-3 polyunsaturated fatty acids (PUFA), eicosapentaenoic acid (EPA) and docosahexaenoic acid (DHA). Studies carried out in a variety of populations including cancer patients,[14] patients with end-stage renal disease,[15] chronic obstructive pulmonary disease[16] and rheumatoid arthritis[17] have shown that dietary PUFAs have a beneficial effect on skeletal muscle mass and strength.

Dietary supplementation of omega-3 PUFAs is of particular interest in the context of frailty, given its well-known anti-inflammatory role and the importance of inflammation in the development of ageing.[18]

Omega-3 reduces inflammation in conditions including Duchenne muscular dystrophy,[19] Crohn's disease,[20] non-alcoholic fatty liver disease,[21] cardiovascular disease[22] as well as many cancers.[14 23–25] These studies are of particular interest, as increased levels of inflammatory biomarkers such as interleukin-6 (IL-6), tumour necrosis factor-alpha (TNF-α) and C reactive protein (CRP) have all been linked with both frailty and sarcopenia in older adults.[26–28] Long-chain PUFAs have been suggested to interact with antioxidants and improve inflammatory responses to positively impact on physical performance.[29] Part of the mechanism may involve the effect of omega-3 on musculoskeletal pain; a pain reduction would be conducive to more physical activity.[30] More generally, omega-3 supplementation may act directly on skeletal muscle and improve protein metabolism hence having an influence on physical performance.[31 32] A more proinflammatory diet has also been linked with a higher incidence of frailty.[33]

A recent review by Ticinesi *et al*[18] summarised the analysis of omega-3 PUFAs on inflammation in older individuals in both cross-sectional and randomised controlled trials (RCTs). However, we are not aware of any systematic reviews or meta-analyses focusing specifically on omega-3 fatty acid supplementation on frailty phenotypes in older adults not selected for any specific chronic or acute conditions. We propose to conduct a systematic review and meta-analysis to examine the effect of dietary omega-3 supplementation in older people not recruited for any given disease diagnosis. The outcomes that will be investigated are inflammatory biomarkers, muscle mass, physical activity, walking speed, weight change, hand grip strength or muscle strength, as well as biases in the included studies.

## METHODS AND ANALYSIS
### Registration
This protocol has been registered with PROSPERO (registration number CRD42017080240)[34] and reported in accordance with Preferred Reporting Items for Systematic Reviews and Meta-Analyses (PRISMA)[35] and PRISMA-Protocol[36 37] guidelines.

### Study selection criteria
#### Interventions and population
Studies reporting results of interventions involving dietary supplementation of omega-3 PUFAs will be included. Dietary supplementation will be defined as daily ingestion of capsules containing EPA and DHA or through an EPA and DHA-enriched diet. The comparator will be placebo-controlled groups. Participants will include community-dwelling persons, of either sex, classified by the study authors as postmenopausal or older people with the majority of participants over 60 years of age. To ensure the focus of the review is on older people not recruited for any given disease diagnosis, exclusions will be studies where participants were selected because they had a cancer or other chronic disease diagnosis. Participants who currently consumed a high fish diet or use fish oil supplements will also be excluded.

#### Outcomes
Outcomes must include changes from baseline to last available follow-up for one or more of the following: muscle mass, inflammatory biomarkers, physical activity, walking speed, weight change, hand grip strength or muscle strength. Any adverse effects will also be summarised. Studies will be grouped for meta-analysis according to the outcome(s) provided.

#### Study designs
RCTs will be included.

#### Other
Articles up until 31 December 2017 in any language will be included.

#### Exclusion criteria
Studies will be excluded for the following reasons: (1) study population was specifically focused on participants diagnosed and being treated for a pre-existing medical condition (eg, cancer, kidney disease, liver disease, diabetes mellitus and cardiovascular disease); or (2) letters to the editor, meta-analyses, case reports and reviews.

#### Search strategy
MEDLINE (Ovid) from 1946, the Cochrane Register of Controlled Trials (CENTRAL) from 1940, Embase from 1946, Cumulative Index to Nursing and Allied Health Literature from 1937, Allied and Complementary Medicine Database and Web of Science will be searched for relevant trials. The search strategy for Medline has been developed in consultation with a subject-specific librarian

and will be adapted for use in other databases. Search terms are informed by Cochrane Handbook[38] and other systematic reviews investigating PUFA dietary supplementation, sarcopenia and/or frailty.

The full search strategy can be found in the online supplementary file 1. Example of searches that will be used can be seen in online supplementary file 2, box 1 MEDLINE (OVID) Advanced Search Example. Syntax (truncation, wildcards and quotation marks) and operators will be amended according to the specific databases. Initial search results will be uploaded to EndNote X7 (Thomas Reuters) prior to the review of titles and abstracts.

## Data extraction

Initial title and abstract review will be conducted by the first author (JS). Duplicates and articles clearly not meeting the selection criteria will be removed. The reference lists from identified letters to the editor, meta-analyses, case reports and reviews will be scanned to identify further trials. Two independent researchers (JS and AMV) will then read the full text of remaining relevant papers for eligibility. In cases where the two researchers cannot agree on eligibility, a third researcher will mediate. Authors of grey literature will be contacted when conference abstracts and proceedings are found. A PRISMA flow chart will be used to provide transparency of the number of papers included or excluded at each stage. Two independent researchers (JS and AMV) will extract the data. The data extracted from the studies (if available) will include (1) authors; (2) publication year; (3) country; (4) funding; (5) setting; (6) study design; (7) sample size; (8) dosage; (9) duration of monitoring or intervention; (10) withdrawals; (11) mean age; (12) gender; (13) muscle mass; (14) physical activity; (15) muscle strength; (16) walking speed; (17) weight; (18) hand grip strength; and (19) biomarkers.

## Risk of bias assessment

Reporting bias will be assessed by plotting the inverse of the SEs of the effect estimates using funnel plots where meta-analysis includes more than 10 trials and will be assessed visually for asymmetry[39] and with the Egger's regression test for continuous variables.[40] Analysis will be conducted on Review Manager Software.[41]

JS and AMV will independently assess the risk of study bias using the Cochrane Collaboration's tool for assessing risk of bias in randomised trials.[42] The Cochrane risk of bias tool for RCTs consists of the following seven items: (1) random sequence generation; (2) allocation concealment; (3) blinding of participants and personnel; (4) blinding of outcome assessment; (5) incomplete outcome data; (6) selective reporting; (7) other sources of bias. Questions are rated as having a high, low or unclear level of bias across the seven domains.

## Strength of evidence evaluation

Strength of evidence will be assessed by Grades of Recommendation Assessment, Development and Evaluation (GRADE) system,[43] that is, quality of evidence for each outcome, relative importance of outcomes and overall quality of evidence.

## Data management and statistical analysis

Data obtained through data extraction will be entered into Excel. Outcomes will be imported into RevMan[41] for meta-analysis. Data extracted must be presented as mean and SD, not ranges, and will not be estimated from graphs or figures. Authors will be contacted if mean and SD values are not presented.

We will create a table describing study characteristics and major outcomes. We will provide a narrative synthesis of the findings from the included studies, structured around the type and content of intervention (ie, diet alone or in combination with training), target population characteristics (ie, sex, age and body mass index (BMI)), type of outcome (ie, muscle strength, physical performance, muscle mass and cognitive function). We will provide summaries of intervention effects for each study by calculating risk ratios (for dichotomous outcomes) or standardised mean differences (for continuous outcomes).

We anticipate that there will be limited scope for meta-analysis because of the range of different outcomes measured across the small number of existing trials. However, where studies have used the same type of intervention, with the same outcome measure, we will pool the results using a random effects meta-analysis, with standardised mean differences for continuous outcomes and risk ratios for binary outcomes, and calculate 95% CI and two-sided p values for each outcome. In studies where the effects of clustering have not been taken into account, we will adjust the SD for the design effect. Heterogeneity between the studies in effect measures will be assessed using the $I^2$ statistic.[44] We will consider an $I^2$ value greater than 50% indicative of moderate heterogeneity or 75% high heterogeneity.[45]

We will conduct sensitivity analyses based on study quality. We will use stratified meta-analyses to explore heterogeneity in effect estimates according to participant characteristics (eg, sex, age and BMI), location (eg, hospital or community setting), intervention components (eg, diet alone or in combination with training) and the logistics of intervention provision.

## Outcomes and prioritisation

The primary outcomes will include recognised frailty criteria of physical activity levels, walking speed, hand grip strength or muscle strength and weight[4] along with changes in muscle mass and circulating levels of the proinflammatory markers CRP, IL-6 and TNF-α.

Other outcomes will be analysed if available including body fat mass.

## Patient and public involvement

The research question was developed following a patient involvement event in May 2017 with members from Arthritis Research UK Pain Centre and National Institute

for Health Research Biomedical Research Centre Musculoskeletal PPI group. Members of the PPI group informed the authors that they already take a number of pharmaceutical treatments and they have a preference for learning more about possible lifestyle interventions, such as dietary modification where they can take control of their only health and well-being. Patients were not involved in the design of this systematic review.

## ETHICS AND DISSEMINATION

No research ethics approval is required for this systematic review, as no confidential patient data will be used. It is intended that the results of this systematic review will be disseminated through publication in an open-access peer-reviewed journal and through conference presentations. All amendments to the protocol will be documented, dated and reported in the PROSPERO trial registry.

## DISCUSSION

This systematic review will use rigorous methodology to identify and examine studies reporting the outcome of omega-3 supplementation on frailty-related traits on ageing groups not selected for specific chronic or acute conditions, including both inflammatory biomarkers and functional measures. No systematic review has previously addressed this objective, although numerous published reviews have focused on ageing populations suffering from chronic or acute conditions. For example, there is evidence of beneficial effects of omega-3 supplementation for individuals undergoing chemotherapy or radiotherapy for cancer[46] for risk reduction in individuals with established atherosclerotic cardiovascular disease[47] and some beneficial effect on liver function in individuals with non-alcoholic fatty liver disease[48] among others.

Although risk of bias and overall level of evidence may limit analyses and confidence in this review's conclusions, this synthesis will provide a better understanding of the effect of omega-3 supplementation in preventing systemic inflammation and functional decline in the elderly population.

### Implications of results

This review will provide the first rigorous summary of effect of omega-3 supplementation across all published randomised controlled trial studies of elderly individuals not selected for chronic or acute conditions. The findings will inform our understanding of the value of this popular nutritional supplement in preventing frailty-related outcomes.

**Acknowledgements** The authors would like to thank Professor Jo Leonardi-Bee for her critical review of the manuscript.

**Contributors** JS, the guarantor of the protocol, drafted the protocol and registered it in PROSPERO. Both authors drafted the manuscript and contributed to the development of the selection criteria, the risk of bias assessment strategy, data extraction criteria and search strategy. All authors read, provided feedback and approved the final manuscript.

**Funding** This work was supported by Arthritis Research UK (grant number 18769) and National Institute for Health Research Nottingham Biomedical Research Centre.

**Competing interests** None declared.

**Patient consent** Not required.

**Provenance and peer review** Not commissioned; externally peer reviewed.

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
