## [Reviewer comments · BMJ Open]

ARTICLE DETAILS

TITLE (PROVISIONAL)	Effect of dietary omega-3 fatty acid supplementation on frailty related phenotypes in older adults: a systematic review and meta-analysis protocol.
AUTHORS	Stocks, Joanne; Valdes, A

VERSION 1 – REVIEW

REVIEWER	Michele Callisaya University of Tasmania, Australia
REVIEW RETURNED	29-Jan-2018

GENERAL COMMENTS	The proposed review is likely to provide useful information. However the following should be addressed: 1. The population to be studied wasn't clear (PRISMA Q8). For example the authors argue that medically stable people will be included. This is a very broad range of people which would include many groups that have been studied and that have chronic disease. At other points the authors state they will include healthy older people. However, the inclusion criteria includes those in hospital or nursing homes. Which chronic diseases will be excluded? Most older people will have arthritis, diabetes, hypertension for example.2. How will the review differ from the recently published review in Int J Mol Sci 2018 19 (1)3. The mechanisms of how supplementation might affect physical activity should be mentioned.4. Search terms – will terms for your outcomes be included?5. Page 10 line 11 – weight is included here but not elsewhere6. There are boxes left blank in the PRISMA checklist7. It would be useful to pull out information on dosage for each study8. PRISMA Q17 – how will the strength of the evidence be assessed?9. Page 4 line 44 – grammatical error
---

REVIEWER	Martine Puts, RN PhD Associate Professor Lawrence S. Bloomberg Faculty of Nursing University of Toronto Toronto, Ontario, Canada
REVIEW RETURNED	31-Jan-2018

GENERAL COMMENTS	While the methods of the review are well described in the protocol, I have a question about the inclusion criteria. In the abstract and full text it is mentioned that there have been reviews of these supplements in older adults with acute or chronic diseases so they argue they will do this focusing on medically stable older people.
--

	However, I wonder how medically stable is defined as the definition is not provided, and how many older adult would meet that inclusion criteria as the majority of older adults have 1 or more chronic diseases. Related to that, why was the cut-off of 60 years chosen for older adults, as it is still quite young and no rationale is provided. Furthermore, could you be more specific on the outcomes, how will frailty be defined and included? As defined by each study author? Will studies including just one of these frailty markers be eligible for inclusion? As some of these markers are not typically used as frailty markers such as inflammatory biomarkers. Furthermore, frailty is seen as a multicomponent syndrome and how that is defined in the outcome is not completely clear to me.
--	---

VERSION 1 – AUTHOR RESPONSE

Reviewer: 1. Michele Callisaya

The proposed review is likely to provide useful information. However the following should be addressed:

1. The population to be studied wasn't clear (PRISMA Q8). For example the authors argue that medically stable people will be included. This is a very broad range of people which would include many groups that have been studied and that have chronic disease. At other points the authors state they will include healthy older people. However, the inclusion criteria includes those in hospital or nursing homes. Which chronic diseases will be excluded? Most older people will have arthritis, diabetes, hypertension for example.

Authors' response: We thank the reviewer for highlighting that the study population wasn't clear. Studies which specifically focus on a group of people with a chronic disease will be excluded, for example, studies which investigate Omega 3 supplementation as a treatment for patients undergoing chemotherapy for cancer, recovering from cardiac surgery, patients in ICU, or as an intervention to prevent diabetes. We have now clarified this by referring specifically to individuals not recruited for a specific disease diagnosis (Abstract line 11 and page 5 line 21-22).

2. How will the review differ from the recently published review in Int J Mol Sci 2018 19 (1)

Authors' response: We thank the reviewer for drawing our attention to this article, however this is not a systematic review but an in vitro study investigating the effect of omega 3 supplementation on lipid metabolites. Also, the study is in severely hyperlipidemic patients with atherosclerotic disease and our review would exclude this study from analysis as it is focused on patients with a specific chronic disease.

3. The mechanisms of how supplementation might affect physical activity should be mentioned.

Authors' response: We thank the reviewer for this recommendation and we have included text related to physical performance on Page 4.

4. Search terms – will terms for your outcomes be included?

Authors' response: We thank the reviewer for this suggestion and have included the outcomes in our search terms and in the Medline search example.

5. Page 10 line 11 – weight is included here but not elsewhere

Authors' response: Thank you for alerting us to this. Weight change has now been included as an outcome in the abstract line 15-16, methods page 5 line 29 and data extraction page 7 line 10.

6. There are boxes left blank in the PRISMA checklist

Authors' response: We have now completed box 17.

7. It would be useful to pull out information on dosage for each study

Authors' response: We agree that it would be useful to also extract dosage information from each study and have included that in the method section (page 7 line 8)

8. PRISMA Q17 – how will the strength of the evidence be assessed?

Authors' response: We have now added text indicating that the strength of evidence will be assessed by GRADE system (page 7 lines 26-28)

9. Page 4 line 44 – grammatical error

Authors' response: Unfortunately, we could not find the line 44 on page 4, but we have carefully proof read the last section on page 4 and have made some changes.

Reviewer: 2, Martine Puts

While the methods of the review are well described in the protocol, I have a question about the inclusion criteria.

In the abstract and full text it is mentioned that there have been reviews of these supplements in older adults with acute or chronic diseases so they argue they will do this focusing on medically stable older people. However, I wonder how medically stable is defined as the definition is not provided, and how many older adult would meet that inclusion criteria as the majority of older adults have 1 or more chronic diseases. Related to that, why was the cut-off of 60 years chosen for older adults, as it is still quite young and no rationale is provided.

Authors' response: We thank the reviewer for pointing this out. We have used the term "medically stable" in the sense of not a population selected for a specific diagnosis (i.e. not supplementation for people with cancer or heart disease) as the aim of our study is to investigate the role of aging not associated to any specific pathology. We have changed the terminology through out the paper for "not selected for a given disease". Although most developed world countries have accepted the chronological age of 65 years as a definition of 'elderly' this is a westernized concepts, since we did not want to limit the validity of any studies the UN agreed cut-off is 60+ years to refer to the older population (<http://www.who.int/healthinfo/survey/ageingdefnolder/en/>).

Furthermore, could you be more specific on the outcomes, how will frailty be defined and included? As defined by each study author? Will studies including just one of these frailty markers be eligible for inclusion? As some of these markers are not typically used as frailty markers such as inflammatory biomarkers. Furthermore, frailty is seen as a multicomponent syndrome and how that is defined in the outcome is not completely clear to me.

Authors' response: We appreciate the importance of a clear outcome, which is why have tried consistently to refer to "frailty related phenotypes" namely, walking speed, physical activity, weight change, hand grip strength and muscle strength which constitute the Fried index for frailty, in addition to inflammatory markers. We have corrected any sections of the manuscript where this was not clearly stated.

VERSION 2 – REVIEW

REVIEWER	Michele Callisaya University of Tasmania Australia
REVIEW RETURNED	27-Feb-2018
GENERAL COMMENTS	The authors have addressed most of the concerns. However, the

	other reviewer and myself had concerns regarding the population to be studied and this could still be refined. Can I suggest that your inclusion criteria includes community-dwelling older people and then the exclusion criteria is people with specific acute or chronic diseases. Furthermore the manuscript still contains reference to 'healthy ageing groups' but includes people in hospital and in institutions who would have either acute or chronic disease. I don't think you should mix these three groups. You could either just concentrate on community-dwelling older people at home or in your analyses look at these three groups separately and remove the reference to 'healthy ageing groups'.
--	---

REVIEWER	Martine Puts, RN PhD Associate Professor Lawrence S. Bloomberg Faculty of Nursing, University of Toronto
REVIEW RETURNED	01-Mar-2018

GENERAL COMMENTS	All the comments have been addressed. Good luck with the review!
--

VERSION 2 – AUTHOR RESPONSE

Following the reviewer's suggestion, we have now amended this issue and clarified that we refer to older adults not specifically selected for chronic or acute conditions (marked copy page 3 line 4; 5 line 8; page 9 line 28; page 10 line 13).

We have also removed the inclusion of persons hospitalised or institutionalised (page 5 line 27).

In response to your request for revision to the strength and limitation section we have included that an important strength of the study is the focus on both functional (walking speed, grip strength, get up and go) and inflammatory outcomes (cytokine level) outcomes which will allow to put in context the effect size and direction of effect of omega-3 supplementation and to prioritise outcomes for future RCTs (page 3 lines 5-8).